# Non-Hermitian Holography

Daniel Areán[1,2], Karl Landsteiner[2], Ignacio Salazar Landea[3]

**1** Departamento de Física Teórica, Universidad Autónoma de Madrid, Campus Cantoblanco, 28049 Madrid, Spain
**2** Instituto de Física Teórica UAM/CSIC, c/Nicolás Cabrera 13-15, Campus Cantoblanco, 28049 Madrid, Spain
**3** Instituto de Física La Plata-CONICET & Departamento de Física, Universidad Nacional de La Plata, C.C. 67, 1900, La Plata Argentina

## Abstract

**Quantum theory can be formulated with certain non-Hermitian Hamiltonians. An anti-linear involution, denoted by PT, is a symmetry of such Hamiltonians. In the PT-symmetric regime the non-Hermitian Hamiltonian is related to a Hermitian one by a Hermitian similarity transformation. We extend the concept of non-Hermitian quantum theory to gauge-gravity duality. Non-Hermiticity is introduced via boundary conditions in asymptotically AdS spacetimes. At zero temperature the PT phase transition is identified as the point at which the solutions cease to be real. Surprisingly at finite temperature real black hole solutions can be found well outside the quasi-Hermitian regime. These backgrounds are however unstable to fluctuations which establishes the persistence of the holographic dual of the PT phase transition at finite temperature.**

## 1  Introduction

One of the basic axioms of quantum mechanics is that the dynamics of a quantum system is generated by a Hermitian Hamiltonian. It comes then as a surprise that meaningful

quantum mechanics can be formulated for certain non-Hermitian Hamiltonians, the so-called PT-symmetric quantum mechanics [1,2]. We quickly review the salient features of this PT-symmetric quantum mechanics using a simple example [2]. It will serve as a guideline to construct a non-Hermitian holographic model. Consider the Hamiltonian of a two state system

$$H_{\mathrm{QM}} = \begin{pmatrix} E - i\Gamma & g \\ g & E + i\Gamma \end{pmatrix}. \tag{1}$$

State $A$ is unstable and decays with decay rate $2\Gamma$ whereas state $B$ is also unstable but suffers exponential growth with the same (inverse) rate. Both states can also transform into each other with amplitude $g$. The interpretation of such Hamiltonians is that the physical system under consideration is subject to exactly balanced gain and loss terms with external sources and sinks. Since gain and loss is balanced one expects that it is possible for the system to reach a time independent steady state. Indeed the eigenvalues of the Hamiltonian (1)

$$\epsilon_\pm = E \pm \sqrt{g^2 - \Gamma^2}, \tag{2}$$

are real as long as the interaction is stronger than the gain/loss terms, $|g| > \Gamma$. The gain/loss terms are exchanged by time-reversal $T$, which in quantum mechanics is just complex conjugation. They are also exchanged by the permutation of the subsystems $A$ and $B$ represented by the matrix $P = \begin{pmatrix} 0 & 1 \\ 1 & 0 \end{pmatrix}$. The combined action PT leaves the Hamiltonian invariant. The so called PT-symmetric regime is the one in which the eigenvalues are real. For $|g| < \Gamma$ the eigenvalues come in complex conjugate pairs; this is the PT broken regime, and the transition between the two is known as PT phase transition [2].

Let us now discuss a slightly different aspect of the Hamiltonian (1). As pointed out in [?,3–5] a Hamiltonian in the PT-symmetric regime is related to a Hermitian one by a similarity transformation. In our case we can start from the fact that every Hermitian Hamiltonian acting on a two-state system can be written as

$$H_2 = E\,\mathbf{1} + \vec{g} \cdot \vec{\sigma}. \tag{3}$$

Every two Hamiltonians of this form can be transformed into each other by an $SU(2)$ transformation $D(\vec{\alpha}) = \exp(i\frac{\vec{\alpha}}{2}\vec{\sigma})$ via $H_2' = D^\dagger H_2 D$. For example we start with a Hamiltonian with $\vec{g} = (g', 0, 0)$. An $SU(2)$ transformation generated by $\sigma_2/2$ brings the Hamiltonian into the form

$$H_2' = E\mathbf{1} + g'\sigma_1 \cos(\alpha) - g' \sin(\alpha)\sigma_3. \tag{4}$$

If we now analytically continue to imaginary values of the parameter $\alpha = i\hat{\alpha}$ we find

$$H_{2,\mathrm{nh}} = E\mathbf{1} + g'\sigma_1 \cosh(\hat{\alpha}) - ig' \sinh(\hat{\alpha})\sigma_3. \tag{5}$$

This Hamiltonian is indeed of the form of (1) with $g = g' \cosh(\hat{\alpha})$ and $\Gamma = g' \sinh(\hat{\alpha})$. The restriction $g^2 > \Gamma^2$ is automatically fulfilled. The unitary matrix $D(\alpha)^{-1} = D(\alpha)^\dagger$ becomes the Hermitian one $\eta(\hat{\alpha}) = \eta(\hat{\alpha})^\dagger$ upon the analytic continuation, and $\eta^{-1}(\hat{\alpha}) = \eta(-\hat{\alpha})$. In the regime of real eigenvalues the Hamiltonian (1) is quasi-Hermitian $H_{2,\mathrm{nh}} = \eta(\hat{\alpha})^{-1}H_2\eta(\hat{\alpha})$ [3]. Notice that $H_2$ is invariant under conjugation with $D(\alpha)$ and a compensating rotation of the couplings $\vec{g} \to R(\alpha)\vec{g}$. Hence we can generate the non-Hermitian Hamiltonian from the Hermitian one by transforming the couplings $\vec{g} = (g', 0, 0)$ with

$$\hat{R}(\hat{\alpha}) = \begin{pmatrix} \cosh(\hat{\alpha}) & 0 & i\sinh(\hat{\alpha}) \\ 0 & 0 & 0 \\ -i\sinh(\hat{\alpha}) & 0 & \cosh(\hat{\alpha}) \end{pmatrix}. \tag{6}$$

The case $|g| = \Gamma$ is special. The Hamiltonian is no longer quasi-Hermitian but it can be reached by taking the limit $\hat{\alpha} \to \infty$, $g' \to 0$ while keeping the product fixed. These special values of the couplings are generically known as "exceptional points".

The guiding principle for constructing the holographic model will be to select an operator that transforms in a unitary representation of a continuous compact Lie group. We also introduce classical couplings transforming in the conjugate representation. The transformation to the non-Hermitian theory is implemented via a complexified group element that acts as a similarity transformation on the Hamiltonian. A typical example in field theory is the Dirac mass term $\bar{\Psi}\Psi$ and the axial mass term $i\bar{\Psi}\gamma_5\Psi$. These transform into each other by axial phase rotations on the Dirac spinor. Starting from the usual mass term and doing a complexified axial transformation one generates the non-Hermitian operator $\bar{\Psi}\gamma_5\Psi$ [6–9][1] Once the quasi-Hermitian theory is obtained it can be extended to the exceptional point and beyond.

## 2 Holography

Gravitational theories with a negative cosmological constant and asymptotically anti-de Sitter boundary conditions allow for a dual interpretation in terms of strongly coupled quantum systems [10]. This can be used to construct gravity models that are dual to interesting quantum many body phenomena [11, 12]. We will now construct the holographic dual to a non-Hermitian quantum field theory along the same lines as outlined before. The key is that in the holographic duality the asymptotic values of the fields encode the couplings of the dual field theory.

In gauge-gravity duality every global symmetry of the dual field theory is promoted to a gauge symmetry in the bulk. To copy our construction for non-Hermitian theories we therefore need at least a $U(1)$ gauge symmetry. In order to introduce couplings transforming under this symmetry we also need a charged bulk field. We simply choose a complex scalar field in the bulk with charge $q$ under the $U(1)$ symmetry. These are the minimal ingredients to construct our non-Hermitian holographic model. Its action

$$\mathcal{S} = \int \sqrt{-g}\, d^{d+1}x \left[ R - 2\Lambda - |D\phi|^2 - m^2|\phi|^2 - \frac{v}{2}|\phi|^4 - -\frac{1}{4}F_{ab}F^{ab} \right] \tag{7}$$

is that of the holographic superconductor [13]. The quartic potential is needed for the model to have domain wall solutions interpolating between two AdS geometries.

For concreteness we will from now on choose $d = 3$ corresponding to the spacetime dimensions of the dual field theory. Furthermore we set $\Lambda = -d(d-1)/(2L^2)$. The equations

---

[1]In the full quantum theory the effects of the axial anomaly should also be accounted for. This is however outside the scope of the present work.

of motion are

$$R_{ab} + g_{ab} \left[ \frac{F^2}{8} + \frac{m^2}{2}|\phi|^2 + \frac{v}{4}|\phi|^4 + \frac{1}{2}|D\phi|^2 - \frac{R}{2} - 3 \right] =$$

$$+ \frac{1}{2} F_{ac} F_b^c + \frac{1}{2} \left( D_a\phi \, \bar{D}_b\bar{\phi} + D_b\phi \, \bar{D}_a\bar{\phi} \right) , \tag{8a}$$

$$\frac{1}{\sqrt{-g}} \partial_a \left( \sqrt{-g} \, F^{ab} \right) - 2q^2 A^b \, \bar{\phi}\phi + iq \, \phi \overleftrightarrow{\partial^b} \bar{\phi} = 0 , \tag{8b}$$

$$\partial_a \left( \sqrt{-g} \bar{D}^a \bar{\phi} \right) + iq \, A_a \, \bar{D}^a \bar{\phi} - m^2 \bar{\phi} - v\bar{\phi}|\phi|^2 = 0 , \tag{8c}$$

$$\partial_a \left( \sqrt{-g} D^a \phi \right) - iq \, A_a \, D^a \phi - m^2 \phi - v\phi|\phi|^2 = 0 , \tag{8d}$$

where $D_a = \partial_a - iqA_a$ and $\phi \overleftrightarrow{\partial^b} \bar{\phi} = \phi \partial^b \bar{\phi} - \bar{\phi} \partial^b \phi$. The unperturbed theory is defined by choosing the asymptotics of the metric. We assume coordinates in which the metric takes the form

$$ds^2 = \frac{1}{z^2} \left[ -u(z)e^{-\chi(z)}dt^2 + \frac{dz^2}{u(z)} + (d\mathbf{x}^2) \right] , \tag{9}$$

and demand that for small values of $z$

$$u(z) = 1 + O(z^2) , \qquad \chi(z) = 0 + O(z^2) , \tag{10}$$

so that it becomes $AdS_4$ as $z \to 0$, and accordingly the conformal boundary is $z \to 0$.

To implement the non-Hermiticity we proceed in the following manner. First we choose general boundary conditions $\phi \approx \exp(i\alpha)\tilde{M}z^\Delta$ and $\bar{\phi} \sim \exp(-i\alpha)\tilde{M}z^\Delta$ where $d - \Delta$ is the conformal dimension of the dual operator determined by the AdS bulk mass through $\Delta = \frac{1}{2}(d - \sqrt{d^2 + 4m^2L^2})$, and for simplicity we take $\tilde{M}$ to be real. Henceforth we set the bulk scalar mass to be $m^2 = -\frac{2}{L^2}$ such that $\Delta = 1$ and set $L = 1$. For the numerical solutions we choose $v = 3/2$ and $q = 1$. Notice that, unlike in the holographic superconductor [14], we explicitly break the $U(1)$ symmetry by the boundary conditions and we do not introduce a chemical potential. Next we promote the theory to a non-Hermitian one by analytically continuing $\alpha \to i\hat{\alpha}$. We also set $e^{\hat{\alpha}} = \sqrt{\frac{1+x}{1-x}}$, $\tilde{M} = \sqrt{1 - x^2}\, M$ and thus obtain the non-Hermitian boundary conditions

$$\phi(z) = (1 - x)Mz + O(z^2) ,$$
$$\bar{\phi}(z) = (1 + x)Mz + O(z^2) . \tag{11}$$

Notice that for $x \neq 0$, $\phi(z)$ and $\bar{\phi}(z)$ are no longer complex conjugate to each other. Let us work out how the PT symmetry acts in our holographic model. We have three fields, the metric, the gauge field and the scalar field. Time reversal acts as $t \to -t$ and as complex conjugation on the imaginary unit $i \to -i$. In addition time reversal has an explicit or external action on the fields as follows. For the gauge field it is simplest to write the gauge field as one-form $A = A_a dx^a$, similarly the metric can be studied via the line element $ds^2 = g_{ab}\, dx^a dx^b$. Time reversal acts as $A \to -A$, $ds^2 \to ds^2$ and $\phi \leftrightarrow \bar{\phi}$. Parity flips the sign of one spatial boundary coordinate $(z, t, x^1, x^2) \to (z, t, -x^1, x^2)$, $A \to -A$ and $ds^2 \to ds^2$. To define the action on the scalar field it is best to write it in terms of real and imaginary parts $\phi = \phi_R + i\phi_I$. Under parity the real part is invariant, whereas the imaginary part is a pseudoscalar and changes sign under parity. The boundary conditions mean that a non-Hermitian operator is sourced in the deformed theory. One way of understanding this is to note that the operator

sourced by $\phi_I$ is an Hermitian operator. Formally, the non-Hermitian operator is sourced by analytically continuing the non-normalizable mode of the real field $\phi_I$ to purey imaginary values. The boundary condition can be written as $\phi_I(z) \to i\tilde{\phi}_I(z) = ixMz + O(z^2)$. Since the external action of T on $\phi_I$ is trivial it follows that the non-Hermitian source field $\tilde{\phi}_I$ changes sign under T. Furthermore $\tilde{\phi}_I$ is a pseudoscalar under parity as is its Hermitian counterpart $\phi_I$. Since non-Hermitian operators are sourced only in the scalar field sector there is no such non-trivial external action of T in the gauge field or metric sector. The Hamiltonian of the dual theory is encoded in the boundary conditions. With the action of T and P we find that the boundary conditions effectively transform with $x \to -x$ under both time reversal and parity. They are thus left invariant under the combined action of PT.

The Hamiltonian of the theory defined by the boundary conditions (11) is indeed PT-invariant. However, it will turn out that the solutions are not PT-invariant when $|x| > 1$. In particular, we will show that the energy of those solutions becomes complex for $|x| > 1$. This allows us to identify $|x| > 1$ as the PT-broken regime.

For $|x| < 1$ our system is in the PT-symmetric phase. In this regime one can easily prove that the solutions of our model are real. Note that the action (7) is invariant under global complexified $U(1)$ transformations, $\phi \to e^{\hat{\alpha}}\phi$ and $\bar{\phi} \to e^{-\hat{\alpha}}\bar{\phi}$. This means that automatically any bulk geometry with non-Hermitian boundary conditions will be the same as an Hermitian one with boundary conditions

$$\phi(z) = \sqrt{1 - x^2}\, Mz + O(z^2)\,,$$
$$\bar{\phi}(z) = \sqrt{1 - x^2}\, Mz + O(z^2)\,. \tag{12}$$

Equivalence of the non-Hermitian and Hermitian theories in the exactly PT-symmetric regime has also been argued for in quantum theory in [15]. Finally $|x| = 1$ is the exceptional point and we comment more on it below.

It is interesting to see explicitly what happens at the border of the quasi-Hermitian regime in holography. To do so we look for solutions with non-Hermitian boundary values. We take the ansatz

$$\phi(z) = (1 - x)\,\psi(z)\,, \qquad \bar{\phi}(z) = (1 + x)\,\psi(z)\,, \tag{13}$$

so that the asymptotic behavior for this new fields reads $\psi \sim Mz + \langle O \rangle z^2$, where $\langle O \rangle$ corresponds to the vev of the dual operator. To find the background we take $\psi(z)$ to be real. Notice that the gauge symmetry in the bulk gives rise to the constraint $\phi\bar{\phi}' - \phi'\bar{\phi} = 0$ which is solved by our ansatz. Finally, the equations of motion (8) boil down to

$$\psi'' + \left(\frac{u'}{u} - \frac{2}{z} - \frac{\chi'}{2}\right)\psi' - (1 - x^2)\frac{2v}{z^2 u}\psi^3 + \frac{2}{z^2\,u}\psi = 0\,,$$
$$\frac{u'}{u} + 3\frac{1 - u}{z\,u} + (1 - x^2)\left[\frac{\psi^2}{z\,u} - \frac{z}{2}\psi'^2 - v\frac{(1 - x^2)}{2z\,u}\psi^4\right] = 0\,,$$
$$\chi' - z(1 - x^2)\psi'^2 = 0\,. \tag{14}$$

## 2.1   $T = 0$ solutions

We will now look for zero temperature solutions that correspond to domain wall geometries. We integrate numerically the equations (14) from a regular solution in the deep IR at large

$z$ [13]

$$u(z) = 1 + \frac{1}{6v} + \dots, \qquad \chi(z) = \chi_0 + \dots, \tag{15}$$

$$\psi(z) = \frac{1}{\sqrt{v}\,\sqrt{1-x^2}} + \psi_1 z^{\frac{3+18v-\sqrt{3}\sqrt{3+68v+300v^2}}{2(1+6v)}} + \dots,$$

which asymptotes to $AdS_4$ with radius $\sqrt{6v/(1+6v)}$ realizing a conformal IR fixed point in the dual theory. $\chi_0$ and $\psi_1$ are two free parameters we use to shoot towards the desired boundary conditions in the UV. The resulting solutions are domain wall geometries interpolating between two $AdS_4$ spaces.

The IR boundary conditions (15) make clear that real solutions can only exist for $|x| \le 1$. For $|x| > 1$ the ground state spontaneously breaks PT and, as we will see, the dual bulk geometry becomes complex.

In figure 1 we show numerical solutions for several values of $|x| < 1$. Since $M$ is the only dimension-full scale, all solutions at fixed $x$ with $M \ne 0$ are equivalent. Then we can explore the space of solutions by simply fixing $M = 1$ and searching for domain walls at different values of $0 \le x \le 1$. We find that the domain wall shifts towards the IR as $x$ is increased,

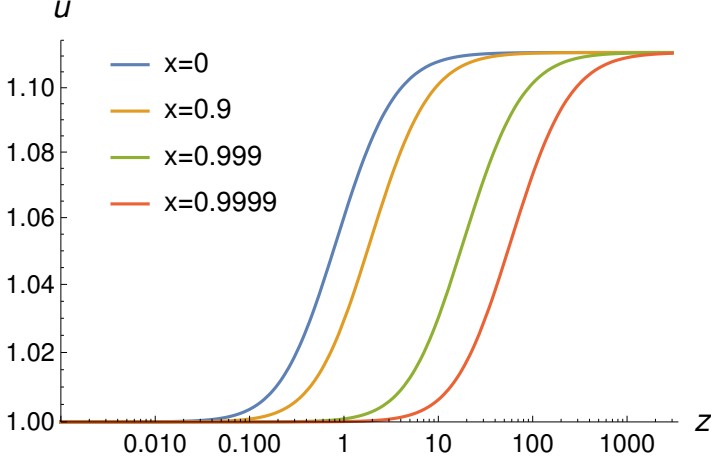

Figure 1: Zero temperature solutions: plot of the metric function $u(z)$ for several values of $x$. As $x \to 1$ the domain wall moves towards the IR ($z \to \infty$).

and in the limit $x \to 1$ it moves all the way to infinity. Indeed, as is clear from (14), at $x = 1$ the scalar decouples from the metric which becomes $AdS_4$, while $\psi = M\,z + \langle O \rangle\,z^2$ is now an exact solution corresponding to a scalar with $m^2 = -2$ in $AdS_4$. Finally, for $|x| > 1$ we find solutions that are complex along the bulk while still meeting the real UV boundary condition $\psi(z) \sim Mz$. In particular, for each value of $x$ we obtain a pair of solutions complex conjugate to each other and featuring a purely imaginary vev $\langle O \rangle$. In figure 2 we plot the free energy of the $T = 0$ solutions around $x = 1$. It can be read from the renormalized on-shell action as $\Omega = -S_{\text{os}} = u_3/2$, where $u_3$ is the subleading contribution of $u(z)$ towards the boundary $u = 1 + u_3\,z^3 + O(z^4)$. We leave the investigation of these complex solutions for future study but note that similar complex solutions have been discussed recently in a different context [16, 17].

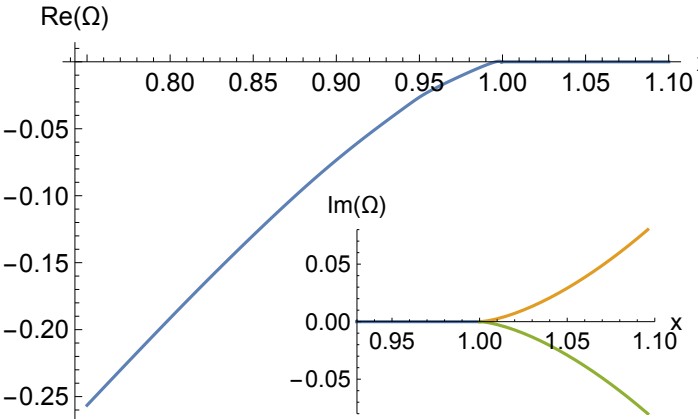

Figure 2: Free energy of the zero temperature solutions as a function of $x$. In the inset we plot the imaginary part, which is nonzero for $x > 1$. We have set $M = 1$.

## 2.2 $T > 0$ solutions

To determine what happens as we heat up the system we now study solutions with an horizon at $z = z_h$ where the blackening factor $u(z_h) = 0$ and

$$\psi(z) = \psi_h - \frac{e^{-\chi_h/2}\psi_h(2 + 3(x^2 - 1)\psi_h^2)}{4\pi z_h^2 T}(z_h - z) + \cdots ,$$

$$\chi(z) = \chi_h + \frac{e^{-\chi_h}(x^2 - 1)\psi_h^2(2 + 3(x^2 - 1)\psi_h^2)^2}{16\pi^2 z_h^3 T^2}(z_h - z) + \cdots ,$$

$$u(z) = 4\pi e^{\chi_h/2} T(z_h - z) + \cdots , \tag{16}$$

with

$$T = \frac{e^{-\chi_h/2}}{16\pi \, z_h} \left[ 12 + (1 - x^2)\, \psi_h^2 (4 - 3(1 - x^2)\, \psi_h^2) \right] \tag{17}$$

the horizon temperature.

Integrating from the horizon and imposing the same UV boundary conditions we now expect a family of solutions characterized by two dimensionless parameters $M/T$ and $x$. Interestingly, we find that at fixed $M/T$ we are able to obtain real solutions for $0 \leq x \leq x_c$, with $x_c > 1$ and monotonically increasing with $M/T$. In figure 3 we plot the vev $\langle O \rangle / M^2$ as a function of $x$ for different values of $M/T$. Notice that for $1 < x < x_c$ two different branches of solutions exist. Finally, beyond $x_c$ we only find complex solutions (with real values of $M/T$).

How is it that we are finding a seemingly valid background of the theory in the PT-broken regime $1 < x \leq x_c$? As we will show next, these solutions have a tachyon in their spectrum and are therefore unstable.

In order to assess the stability of our finite temperature solutions we now study the quasi-normal modes (QNM) of the system. More precisely we look for solutions to the spacetime-dependent linearized equations of motion with ingoing boundary conditions at the black hole horizon. These fluctuations can be organized in several decoupled sectors. We focus on the one containing the following components of the gauge field $\delta A = e^{-i\omega t + ikx^1}(a_t(z)dt + a_1(z)dx^1)$ and a particular combination of the fluctuations of the scalar fields defined through the constraint

$$e^{-i\omega t + ikx^1}(1 - x)\,\delta\bar{\phi}(z) = e^{-i\omega t + ikx^1}(1 + x)\,\delta\phi(z) . \tag{18}$$

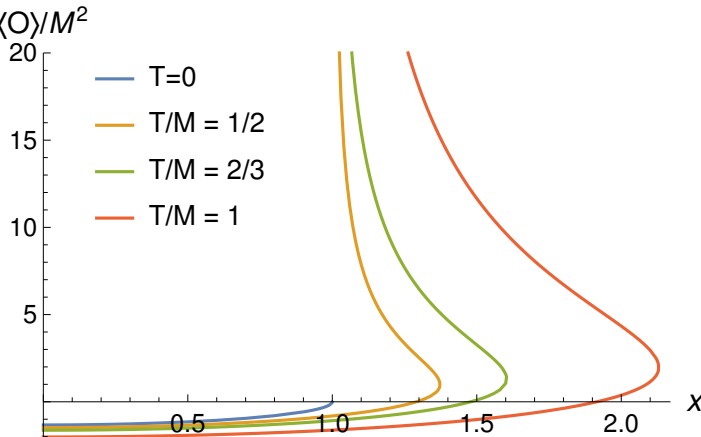

Figure 3: Finite temperature solutions: plot of the vev as a function of $x$ for different values of $T/M$. The $T = 0$ result is included for comparison.

This constraint results from requiring that the Einstein's equations of motion are satisfied without turning on any new metric degree of freedom. Solving (18) for $\delta\phi$ when $x > 0$ (analogously one solves for $\delta\bar\phi$ when $x < 0$), the equations of motion for $\delta\phi$ and $\delta\bar\phi$ become equivalent, and we are left with the following three coupled differential equations

$$
\delta\bar\phi'' + \left[\frac{u'}{u} - \frac{1}{2z}(4 + z\chi')\right]\delta\bar\phi' + (x+1)\,q\,\omega\,e^\chi\frac{\psi}{u^2}\,a_t + (1+x)\frac{\psi}{u}\,q\,k\,a_1
$$
$$
- \left[\omega^2\frac{e^\chi}{u^2} - \frac{1}{uz^2}\left(m^2 + (x^2-1)\frac{v}{2}\psi^2\right)\right]\delta\bar\phi = 0\,,
$$
$$
a_1'' + \left[\frac{3}{ru} - \frac{3}{r} - \frac{(x^2-1)\psi^2}{ru} - \frac{(x^2-1)\,v\,\psi^4}{4ru}\right]a_1' + \left[\frac{(x^2-1)q^2\psi^2}{r^2u} + \frac{e^\chi\omega^2}{u^2}\right]a_1
$$
$$
- \frac{(x-1)2k\,q\,\delta\bar\phi}{r^2u} + \frac{e^\chi k\,\omega\,a_t}{u^2} = 0\,.
$$
$$
\omega\,z^2 a_t' + e^{-\chi}u\,k\,a_1' + 2q\,e^{-\chi}(1-x)\,u\,(\psi\delta\bar\phi' - \psi'\delta\bar\phi) = 0 \tag{19}
$$

We integrate these equations numerically, imposing ingoing boundary conditions at the horizon

$$
\delta\bar\phi(z) = (z_h - z)^{-\frac{i\omega}{4\pi T}}\left[\delta\phi_h + O(z_h - z)\right],
$$
$$
a_t(z) = (z_h - z)^{-\frac{i\omega}{4\pi T}+1}\left[\frac{8\pi qe^{-\chi_h/2}(x-1)T\psi_h\delta\phi_h}{z_h^2(4\pi iT + \omega)} + O(z_h - z)\right],
$$
$$
a_1(z) = (z_h - z)^{-\frac{i\omega}{4\pi T}}\left[a_{1h} + O(z_h - z)\right], \tag{20}
$$

which corresponds to the computation of the retarded Green's function. We will be interested in its lowest lying poles.

Since we do not know how to decouple these equations of motion (19) we will use the determinant method to compute them [18].. This means that we will build a $3 \times 3$ matrix with the leading UV values for our perturbations for two linearly independent solutions. From (20) we see that we only have two free parameters at the horizon. In order to make our method

work we include the pure gauge solution

$$a_t(z) = -\omega\,, \quad a_1(z) = k\,, \quad \delta\bar{\phi}(z) = q(1+x)\psi(z)\,. \tag{21}$$

The zeroes of the determinant of the matrix of solutions evaluated at the boundary correspond to poles in the Green's function in the mass basis and we can easily find them by integrating (19) from the horizon towards the UV and using $\omega$ as our shooting parameter. We also note that the system of equations degenerates to rank two in the case of zero momentum $k = 0$.

In figure 4 we plot, for $k = 0$, the purely imaginary QNM that becomes the pseudo-diffusive one at $x = 0$. This is the would-be hydrodynamic mode corresponding to charge diffusion. Since the symmetry is broken by the parameter $M$ the mode becomes dissipative (i.e. takes a negative imaginary value) even at $k = 0$. As $x$ is increased the purely imaginary gap decreases, vanishing at exactly $x = 1$. Recall that at $x = 1$ the scalar decouples from the geometry and we recover the hydro diffusive mode. Indeed the scalar field fluctuations decouple from the gauge field fluctuations in (19). These gauge field fluctuations in AdS generically have a diffusive mode in the quasi-normal mode spectrum [19]. Another way to see this is by considering $x \lesssim 1$. Then one can perform the complexified gauge transformation to go to the Hermitian theory with vanishing boundary conditions. This implies that the profile for the scalar is $\phi(z) = 0$ in the limit $x \to 1$. Looking at the corresponding pure gauge solutions one finds that one is simply left with $a_t(z) = \omega$ signaling the presence of a pole precisely at $\omega = 0$.

Crucially, for $x > 1$ the mode crosses into the upper half plane, thus becoming tachyonic and signaling the instability of those finite temperature solutions beyond the PT-symmetric regime. One could ask next if this instability leads to a new background for $x > 1$. For the system at hand the only possibility would be that a background with a spontaneous nonzero charge density $(A_t(z) = -\rho\, z + \dots)$ exists for $x > 1$. Yet a thorough numerical search has failed to produce such a background (even after relaxing the requirement that the fields be real). We thus believe that there is no endpoint for this instability indicating that the system does not have a true ground state in the PT-broken regime.

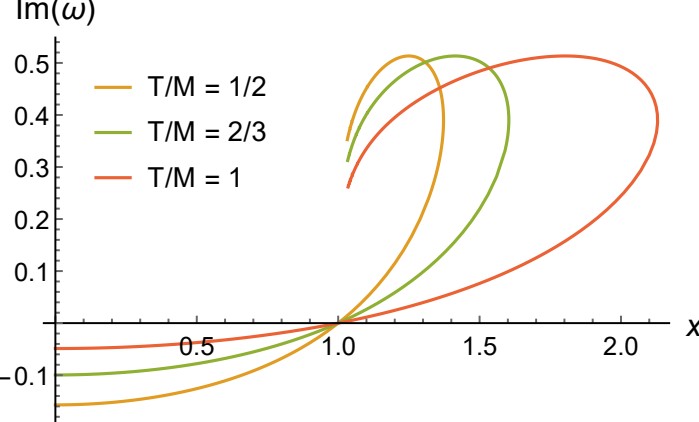

Figure 4: Pseudo-diffusive mode as a function of $x$ for different values of $T/M$.

Let us end our analysis of the QNMs by turning on the spatial momentum $k$. In figure 5 we plot the $k$ dependence of the QNM for several values of $x = 0, 0.5, 1, 1.2$ at fixed $T/M = 1/2$.

The intercepts at $k = 0$ naturally agree with the corresponding values of the gap depicted by the yellow line in figure 4.

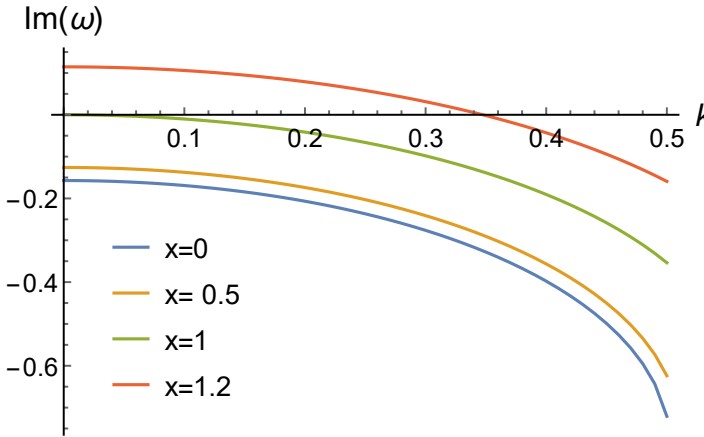

Figure 5: $k$ dependence of the pseudo-diffusive mode for several values of $x$ at $T/M = 1/2$.

## 3    Conclusion and Outlook

We have successfully constructed a model of a strongly coupled quantum system with non-Hermitian couplings via the holographic duality. The PT phase transition takes an interesting form at finite temperature: real solutions exist even for a region of values $|x| > 1$, but they turn out to be unstable to small fluctuations. While our model falls into the bottom-up class it can be easily generalized to models directly derived from string theory such as the ones in [20–22]. We expect our findings to hold also in these models. There are many possible generalizations of our work. Spontaneous symmetry breaking and Goldstone modes in PT field theories have recently been discussed in [23–28]. This could be generalized to holographic systems using the methods of [29–31]. It would also be interesting to understand if a similar picture holds for the PT phase transition at finite temperature in weakly coupled perturbative field theory. Finally we note that gauge/gravity duality with open boundary conditions and decoherence has recently been studied in [32]. It would be interesting to see its relation to the PT-symmetric model presented here.

## Acknowledgements

We thank M. Ammon, A. Amoretti, M. Baggioli, A. Cortijo, M. Chernodub, D. Dubois, C. Hoyos, A. Jiménez, E. Kiritsis, for useful discussions. I.S.L. thanks ICTP, IFT and IB for hospitality during different stages of this project. D.A. is supported by the 'Atracción de Talento' programme (Comunidad de Madrid) under grant 2017-T1/TIC-5258. D.A. and K.L are supported by MCIU/AEI/FEDER, UE, through the grants SEV-2016-0597 and PGC2018-095976-B-C21. I.S.L. is a Conicet fellow.

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
