# Peer review of "Non-Hermitian Holography"

_SciPost Physics Core_

## Round 1 · Referee Report · Anonymous (Referee 1) · 2020-4-10

Report

This paper presents a possible gravity dual realization of a non- Hermitian, ''PT symmetric'' strongly coupled quantum field theory. Non-Hermitian PT symmetric theories have been studied in the context of dissipative and open systems, as well as formal developments of quantum mechanics.

By analogy with a simple quantum mechanical system it is proposed that a holographic non-Hermitian model can be obtained from a Hermitian model by a complexified symmetry transformation. The concrete model the authors present is a complex scalar field coupled to gravity and a Maxwell field, with non-Hermitian boundary conditions for the scalar, which would map to non-Hermitian couplings in the field theory dual. Among the possible boundary conditions, the authors identify a ''quasi-Hermitian'' subset that can be obtained from Hermitian conditions through a complexified global $U(1)$ rotation acting on the complex scalar. They construct zero and finite temperature solutions of the gravity model and find that stable real solutions only exist for quasi-Hermitian boundary conditions.

To my knowledge, there have been no previous works attempting to describe non-Hermitian systems in the context of gauge/gravity duality, this is a new an interesting application and certainly deserves to be published. There is maybe some room for improvement, some minor comments/questions I have are the following:

1) The authors identify the quasi-Hermitian regime with a PT-symmetric regime, and otherwise they mention that PT would be broken. It would be worthwhile to specify exactly what PT transformations are in the holographic model and how they are broken in one case but not the other.

2) In the paragraph below Eq. (20) it looks like the authors are saying that at $x=1$ the lowest QNM they are following becomes the hydrodynamic mode for charge diffusion. This sounds a bit strange, from (19) it looks like at at $x=1$ the fluctuation for the gauge field should be set to zero and the mode is purely a fluctuation of the scalar, that will be decoupled from the metric and the gauge field (the charge of the scalar is effectively zero).

3) For $x>1$ the authors find that real finite temperature solutions become unstable and were not able to find other numerical solutions. I was wondering if a transition of Hawking-Page type would be possible to the thermal version of the zero temperature solutions they found previously. By thermal version I mean that in the analytic continuation to Euclidean signature they would have a compact time direction of finite length, corresponding to the inverse temperature.

4) Some possible typos: below equation (15), in the paragraph starting ''The IR boundary conditions (16)..'' do the authors actually mean the previous equation, (15)? In equation (19) it looks like there is a factor $q$ missing in the first equation in the term proportional to the gauge field.

  • validity: -
  • significance: -
  • originality: -
  • clarity: -
  • formatting: -
  • grammar: -

Author:  Karl Landsteiner  on 2020-04-17  [id 795]

(in reply to Report 1 on 2020-04-10)

We would like to thank the referee for her/his positive review and interesting comments.

Let us address the points mentioned in the report:

1) The holographic image of the PT transformation is complex conjugation. In the PT symmetric regime the solutions are real. In the regime where PT is broken the solutions come in complex conjugate pairs, as demanded by PT symmetry.

More specifically it is interesting to analyse how the PT symmetry acts on the scalar field $\phi$. To this end we write it in terms of its real and imaginary parts $\phi = a + i b$. The real part is a scalar whereas the imaginary part is a pseudo scalar, i.e. under parity: $P: (a, b) \rightarrow (a , -b)$ Time reversal acts as complex conjugation and takes $T: \phi \rightarrow \bar\phi$. In terms of the real and imaginary parts: $T: (a, b) \rightarrow (a, b)$ In order to go to the non-Hermitian theory we take the imaginary part $b \rightarrow -i \tilde b$ . Therefore $T: \tilde b = -\tilde b$

Now it can be seen that the combined action PT leaves the complexified fields $\phi=a+b$ and $\bar \phi = a-b$ invariant but acts as complex conjugation otherwise (e.g. on the metric). Therefore the boundary conditions we impose respect PT symmetry. The solutions we find are trivial representations of PT in the PT-symmetric regime and form complex conjugate pairs in the PT-broken regime. This is precisely as in PT quantum mechanics.

2) Let us emphasize, that, upon plugging in our non-Hermitian ansatz in the action (7), at x=1 the whole scalar sector decouples, and we are left with the Maxwell-Einstein action for the metric and gauge fields. Therefore one is bound to find the charge diffusion mode as we do.

3) We do not think that there is a Hawking-Page type phase transition to a stable background. The Hawking-Page type transition is generically a first order transition whereas here we have an instability due to small, linear fluctuations. The zero temperature solution with the same boundary condition is the one we have already found before and is complex for |x|>1. The equations that have to be solved to find the zero temperature solutions are time independent and therefore not sensitive to changing to Euclidean signature nor are they sensitive to the Euclidean time being compact. Therefore the IR solution (15) seems the only regular one at zero temperature, even when allowing for Euclidean solutions where $t\rightarrow i \tau$. Notice that (15) becomes complex for $x>1$. Moreover, in a Hawking-Page like transition there are typically two compact cycles, e.g. a circle and a sphere in the case of the original Hawking-Page transition, and solutions with two different topologies in which either one or the other compact space shrinks to zero size in the interior of the spacetime. The Euclidean section of our finite temperature solutions in contrast have only one compact cycle, the Euclidean time direction.

4) The referee is right and indeed the reference should be to eq. (15). We think the power of q in equation (19) is correct. One can quickly check this from eq. (8.b) where the corresponding term is the last one.

---

## Round 1 · Referee Report · Anonymous (Referee 1) · 2020-4-20

Report

I want to thaks the authors for their explanations, these are my comments to their reply:

1) I guess my previous comment was not clear enough, I think readers would benefit if a discussion about PT in the holographic model (along the lines in the author's reply) is introduced in the paper. Regarding the reply, in the end I did not understand what the action of T on $(a,b)$ should be when they take complex values. From their comments it looked like they should be invariant, but that would not be consistent with complex conjugation of the metric when this last one takes complex values, as all of them are real in the ordinary setup.

2) Equation (19) does not involve metric fluctuations and when x=1, the only solution for the gauge field is trivial. So it would look like the fluctuation at that point is purely coming from the scalar field and neither the metric nor the gauge field are involved. So I still do not understand the comment about the diffussion mode, or Figure 3 does not represent the solutions of (19)?

3) The first order transition could happen before the system becomes locally unstable, in this case for|x|<1. It is not mandatory that there are two cycles, the relevant topology change concerns only the Euclidean time direction. One may consider a Hawking-Page transition for a planar black hole, for instance the AdS black brane versus the thermal AdS solution, with the thermal AdS solution being the same as the zero temperature solution with a compactified Euclidean time direction, a similar situation could be possible here with the zero temperature solution (15). Whether there is a transition or not would depend on the values of the free energies for each solution, in the AdS case the black brane always has lower free energy, but maybe this case is different.

4) The $q$ factor in In equation (19) I was referring to would be in the term that mixes the background scalar $\psi$ with the gauge field $a_t$ in the first line of the first equation, not the one the authors point out.

  • validity: -
  • significance: -
  • originality: -
  • clarity: -
  • formatting: -
  • grammar: -

Author:  Karl Landsteiner  on 2020-07-13  [id 885]

(in reply to Report 2 on 2020-04-20)

We thank the referee for the pertinent comments and also apologise for the delay in answering. Our answers are as follows:

1) Below (11) we have added an improved discussion on PT in the holographic model.

2) We expanded our discussion about the computation of QNM, making special emphasis on the method we used. Also we turned on finite momentum and we have now explicitly calculated the k-dependence of the mode in question. In this way, we hope to make more why we called it a "pseudo"-diffusive mode that will become tachyonic for $x>1$. In particular the mode in question is purely imaginary, shows quadratic k-dependence but due to the symmetry being broken does not have zero frequency at zero momentum. At x=1 however the equations for the true diffuse QNM mode are recovered due to the decoupling of the scalar field from the gauge field fluctuations at that point. So the appearance of a mode with zero frequency at zero momentum is generic at $x=1$. For $x>1$ this mode moves to the upper half plane and thus represents an instability.

3) We have not been able to find a competing zero temperature solution. If one insists on taking our $x<1$ zero temperature solutions (which are AdS to AdS domain walls) and compactifies the time direction, one of course finds that the size of this compactified direction shrinks to zero at the IR AdS horizon. Also notice that since we are working on the Poincare patch of Ads, for our domain walls the temperature is set to zero by the Poincare horizon. Hence we cannot simply compactify the time direction and compare the free energy with that of a black hole. If the Referee has some ansatz in mind we would be very happy to consider it. Finally we should say that the domain walls we find are in fact simple variants of the ones thoroughly studied in the literature. Specially in the PT symmetric regime where there is a map to those solutions. Hence we do not expect to find any transition that was not previously reported in that regime.

4) The Referee is of course right and we thank her/him for pointing out this typo. We have corrected that and also have included the momentum dependence in eq. (19).

---

## Round 1 · Referee Report · Anonymous (Referee 2) · 2020-5-17

Strengths

A simple idea presented in a short paper that is to the point.

It is not clear where future applications or developments may lead for PT symmetry in holography.

Weaknesses

It is not clear where future applications or developments may lead for PT symmetry in holography.

Report

This paper revolves around the idea of so-called PT symmetric quantum systems. The paper proposes a first realisation of such a system in holography by employing a bulk dual setup well-known from the study of holographic superconductors, albeit with differing boundary conditions.

To the best of my knowledge PT symmetric quantum systems have not previously been considered in holography and, as such I have enjoyed reading this work.

I would like to ask the authors to address the following questions or suggestions:

  1. I have found the way the holographic example of a PT symmetric quantum system is introduced to be a little unclear.

  2. The relationship between the simple QM example given in the introduction and the actual holographic model is not clear. One might invoke, for example, the crucial role of the SU(2) symmetry in the QM system, which is absent in the holographic setup. Could the authors clarify the relationship between the two models further?

  3. In the paragraph before the section “holography” the authors seem to insist on the fact that the Lie group that is used be simple. This does not appear obvious from the examples before and after.

  4. Neither the parameter $\tilde M$ nor the parameter $M$ are defined, nor is their mutual relationship. Are they both real?

  5. It would appear to me that the crux of their setup lies in the analytic continuation of the boundary conditions, and hence the bulk solutions, such that $\bar\phi$ is no longer the complex conjugate of $\phi$. If this understanding of mine is correct, it might be nice to state this somewhere explicitly, it may even obviate the need to transform between various parameters $\tilde M$, $M$ and $x$, which all encode in some form or another this boundary condition.

  6. Some further questions aimed at clarification:

  7. The simple $T=0$ solutions of domain-wall type are strongly tied to the form of the bulk action they chose, which is not fully dictated by their symmetry considerations. What is the motivation for the action (7) in particular? Could the authors comment on whether they expect similar physics for other choices of the potential?

  8. After Eq. (12) the authors invoke the equivalence of non-Hermitian and Hermitian theories in the PT-symmetric regime, citing Ref. [15]. However, this point had already been discussed before, and backed up by Refs. [3-5]. Is there something qualitatively different that I am missing?

  9. Some suggestions / comments:

  10. The authors leave the reader with the puzzling fact that it seems like the finite-temperature solutions they have found can be extended beyond the expected range of $x<1$, but appear to have an instability beyond $x=1$. It would be useful to address such puzzles in a top-down setting (and the authors have helpfully cited some such examples). In those settings one can hope to be more precise on the exact nature of the dual (i.e. boundary theory), and how the PT symmetry and its breaking are realised.

  11. I would have enjoyed seeing a bit more on motivations as to why the authors are constructing PT symmetric systems in holography. They mention some relations to Goldstone mode physics in the conclusions, however it is not clear to me what a holographic model would bring to this study. Perhaps the authors could clarify this, and at the same time elaborate a little more on the more general perspective the wish to take on PT symmetry in holography.

  • validity: high
  • significance: high
  • originality: good
  • clarity: good
  • formatting: perfect
  • grammar: perfect

Author:  Karl Landsteiner  on 2020-07-13  [id 884]

(in reply to Report 3 on 2020-05-17)
Category:
answer to question

We thank the referee for the pertinent comments, questions and suggestions. We also apologise for the long delay in answering!

1.1)
The presence of a global SU(2) symmetry is not essential to the construction of a non-Hermitian theory via the complexified rotation we describe in the intro. As an example, one can achieve this via a U(1) as we briefly describe for the case of a fermionic mass term.

1.2) We have removed the reference to simple Lie groups since we do not believe that plays any role in our argument.

1.3) The parameters $M$ and $\bar M$ are defined via the asymptotic boundary conditions of the
scalar field. Although not crucial for the construction, we have picked them to be real.
To be precise, $\tilde M$ is defined below eq. (10) as the modulus of the leading asymptotics of $\phi$ and $\bar \phi$. Once we apply the complexified rotation, we write the non-Hermitian boundary conditions as (11) in terms of $M = 1/\sqrt{1-x^2} \tilde M$ where
$e^\alpha= \sqrt{(1+x)/(1-x)}$. These relations are now explicitly written down before (11).
Notice that $M$ is real and thus (11) are clearly non-Hermitian boundary conditions.

1.4) The Referee is right; the complexified rotation of the usual Hermitian boundary conditions results in (11) which are clearly non-Hermitian as long as $x\neq 0$. In our construction the non-hemiticity translates into the fact that the bulk scalars are no longer complex conjugate one to the other. Nonetheless this is not equivalent to gave two real scalars since we still have photons in the bulk.
We have added a sentence below (11) stressing that phi and bar phi are not complex conjugate to each other for $x\neq 0$.

2.1) We choose the action (7) because is the simplest setup enjoying a local U(1) symmetry in which one can construct AdS to AdS domain walls. This setup has been thoroughly studied in the context of Holographic Superconductors. Thanks to the potential having a minimum at a nonzero value of the scalar one can find solutions flowing to AdS in the IR.

2.2) The equivalence between non-Hermitian and Hermitian theories is explored, and proved in some cases in [3-5], and finally made explicit in general in ref. [15].

3.1) We thank the referee for the suggestion. In fact we think that even beyond top-down models the behaviour we found, seemingly well-behaved solutions at finite temperature for T>0 should be investigated also in the weak coupling field theoretical approach. In this sense our results obtained in a dual gravity setup give new directions for research even for weakly coupled field theories.

3.2) The motivation was curiosity driven basic research. PT symmetric quantum field field theory is a rather active field of research and we think that it is a valid and interesting question to ask how PT symmetry plays out in holographic models of strongly coupled quantum field theories. As we have pointed out in the previous point, it gave already useful insight and results that might be relevant beyond the holographic setup. The implementation of spontaneous symmetry breaking in PT symmetric field theories is also under investigation, e.g. refs [23]-[28]. In holography spontaneous symmetry breaking leads to Goldstone mode in the QNM spectrum (refs[29]-[31]), so it should be interesting to investigate the properties of goldstone modes in the QNM spectrum of PT symmetric holographic models. Of course this is subject to future research.

---

## Editorial Decision

resubmitted